# Nanoscale dysregulation of collagen structure-function disrupts mechano-homeostasis and mediates pulmonary fibrosis

Mark G Jones[1]*, Orestis G Andriotis[2†], James JW Roberts[3†], Kerry Lunn[3], Victoria J Tear[3], Lucy Cao[4], Kjetil Ask[5], David E Smart[1], Alessandra Bonfanti[6], Peter Johnson[7,8], Aiman Alzetani[1,9], Franco Conforti[1], Regan Doherty[10], Chester Y Lai[1], Benjamin Johnson[11], Konstantinos N Bourdakos[7,8], Sophie V Fletcher[1,9], Ben G Marshall[1,9], Sanjay Jogai[9], Christopher J Brereton[1], Serena J Chee[9,11], Christian H Ottensmeier[11], Patricia Sime[12], Jack Gauldie[5], Martin Kolb[5], Sumeet Mahajan[7,8], Aurelie Fabre[13], Atul Bhaskar[6], Wolfgang Jarolimek[4], Luca Richeldi[1,14], Katherine MA O'Reilly[15,16], Phillip D Monk[3‡], Philipp J Thurner[2‡], Donna E Davies[1,8‡]*

[1]NIHR Southampton Biomedical Research Centre, Clinical and Experimental Sciences, Faculty of Medicine, University of Southampton, Southampton, United Kingdom; [2]Institute for Lightweight Design and Structural Biomechanics, TU Wien, Getreidemarkt, Austria; [3]Synairgen Research Ltd, Southampton, United Kingdom; [4]Pharmaxis Ltd, Frenchs Forest, Australia; [5]Department of Medicine, Firestone Institute for Respiratory Health, McMaster University and The Research Institute of St. Joe's Hamilton, Hamilton, Canada; [6]Aeronautics, Astronautics and Computational Engineering, Faculty of Engineering and the Environment, University of Southampton, Southampton, United Kingdom; [7]Department of Chemistry, Faculty of Natural and Environmental Sciences, University of Southampton, Southampton, United Kingdom; [8]Institute for Life Sciences, University of Southampton, Southampton, United Kingdom; [9]University Hospital Southampton, Southampton, United Kingdom; [10]Biomedical Imaging Unit, Faculty of Medicine, University of Southampton, Southampton, United Kingdom; [11]CRUK and NIHR Experimental Cancer Medicine Centre, Cancer Sciences, Faculty of Medicine, University of Southampton, Southampton, United Kingdom; [12]Division of Pulmonary and Critical Care Medicine, University of Rochester School of Medicine and Dentistry, Rochester, United States; [13]Department of Histopathology, St. Vincent's University Hospital & UCD School of Medicine, University College Dublin, Dublin, Ireland; [14]Università Cattolica del Sacro Cuore, Fondazione Policlinico Universitario Agostino Gemelli IRCCS, Rome, Italy; [15]Mater Misericordiae University Hospital, Dublin, Ireland; [16]School of Medicine and Medical Science, University College Dublin, Dublin, Ireland

*For correspondence:
mark.jones@soton.ac.uk (MGJ);
d.e.davies@soton.ac.uk (DED)

†These authors contributed equally to this work
‡These authors also contributed equally to this work

**Abstract** Matrix stiffening with downstream activation of mechanosensitive pathways is strongly implicated in progressive fibrosis; however, pathologic changes in extracellular matrix (ECM) that initiate mechano-homeostasis dysregulation are not defined in human disease. By integrated multiscale biomechanical and biological analyses of idiopathic pulmonary fibrosis lung tissue, we identify that increased tissue stiffness is a function of dysregulated post-translational collagen

cross-linking rather than any collagen concentration increase whilst at the nanometre-scale collagen fibrils are structurally and functionally abnormal with increased stiffness, reduced swelling ratio, and reduced diameter. In ex vivo and animal models of lung fibrosis, dual inhibition of lysyl oxidase-like (LOXL) 2 and LOXL3 was sufficient to normalise collagen fibrillogenesis, reduce tissue stiffness, and improve lung function in vivo. Thus, in human fibrosis, altered collagen architecture is a key determinant of abnormal ECM structure-function, and inhibition of pyridinoline cross-linking can maintain mechano-homeostasis to limit the self-sustaining effects of ECM on progressive fibrosis.

DOI: https://doi.org/10.7554/eLife.36354.001

## Introduction

Fibrotic diseases are a major cause of morbidity and mortality worldwide and their prevalence is increasing with an ageing population. Within the lung, idiopathic pulmonary fibrosis (IPF) is considered the prototypic chronic progressive fibrotic disease (*Raghu et al., 2011*). Treatment options are limited, and with a median survival of less than 3 years from diagnosis, more effective therapies are urgently needed (*Richeldi et al., 2017*). Whilst the exact mechanisms of progressive lung fibrosis are uncertain, IPF is thought to result from repetitive micro-injuries to the alveolar epithelium promoting fibroblast differentiation into extracellular matrix (ECM)-producing myofibroblasts. Persistent myofibroblast activation results in ECM deposition which eventually destroys normal alveolar architecture and disrupts gas exchange (*Kirk et al., 1986*).

Excessive deposition of fibrillar collagens is considered synonymous with fibrosis. Fibrillar collagens are a major component of lung ECM that form a scaffold to support tissue architecture and are a primary determinant of tissue stiffness (*Senior et al., 1975*; *White, 2015*). During biosynthesis, collagen molecules acquire several post-translational modifications, including lysine hydroxylation and oxidation that are critical to the structure and biological functions of this protein. Oxidative deamination of lysine or hydroxylysine initiates cross-linking reactions that are essential to stabilise the supramolecular assembly of collagen and produce stable collagen fibrils (*Shoulders and Raines, 2009*; *Kadler et al., 1996*). The lysyl oxidase (LOX) enzymes are a family of five secreted copper-dependent amine oxidases (LOX and LOX-like (LOXL) 1 to 4) that are responsible for post-translational modification of collagen in the ECM to initiate covalent cross-linking. LOX family members have been implicated as possible therapeutic targets in cancers and fibrosis (*Trackman, 2016*). The type of LOX/LOXL mediated collagen cross-link is determined by the hydroxylation of telopeptidyl and helical lysine residues in collagen prior to cross-link formation (*Yamauchi and Sricholpech, 2012*), with increased hydroxylation of telopeptide lysine residues by lysyl hydroxylase 2/procollagen lysine,2-oxoglutarate 5-dioxygenase 2 (LH2/PLOD2) proposed to be a general fibrotic phenomenon causing increased hydroxyallysine derived pyridinoline cross-links (*Brinckmann et al., 1999*; *van der Slot et al., 2003*).

Both ECM amount and altered post-translational modifications are postulated to increase matrix stiffness and this stiffening has been proposed to induce self-sustaining mesenchymal cell activation and progressive fibrosis in a positive feedback loop (*Wipff et al., 2007*; *Liu et al., 2010*; *Zhou et al., 2013*; *Booth et al., 2012*; *Chen et al., 2016*; *Parker et al., 2014*; *Shi et al., 2011*; *Liu et al., 2015*). Whilst there is a growing understanding of mechanosensitive cellular pathways that are activated by increased ECM stiffness, the specific changes in ECM structure and function that disrupt mechano-homeostasis have not been defined in human fibrosis (*Burgstaller et al., 2017*; *Tschumperlin et al., 2018*; *Herrera et al., 2018*). To investigate this, we performed the first integrated multiscale structure-function analysis of human fibrosis tissue, and then extended our findings into mechanistic studies in vitro and in vivo. We identify that, at the time of diagnosis, in human lung fibrosis tissue altered collagen architecture rather than collagen concentration is a key determinant of abnormal ECM structure-function, and that targeting pathways which dysregulate collagen architecture may restore ECM homeostasis and so prevent persistent mechanosensitive cellular activation and fibrosis progression.

**eLife digest** Idiopathic pulmonary fibrosis (IPF) is a devastating disease of the lung, which scars the tissue and gradually destroys the organ, ultimately leading to death. It is still unclear what exactly causes this scarring, but it is thought that increasing amounts of proteins in the space surrounding the cells of the lungs, the extracellular matrix, could play a role.

These proteins, including collagen, normally form a 'scaffold' to stabilize cells, but if they accumulate uncontrollably, they can render tissues rigid. It has been assumed that these changes are a consequence of the disease. However, recent evidence suggests that the increased stiffness itself could stimulate cells to produce even more extracellular matrix, driving the progression of the disease. A better understanding of what exactly causes the tissue to become gradually stiffer may identify new ways to block the progression of IPF.

Now, Jones et al. compared measurements of the tissue stiffness and the collagen structure taken from samples of patients with IPF. The results showed that the collagen fibres were faulty and had an abnormal shape. This suggests that these problems, rather than an increased amount of collagen, alter the flexibility of the lung tissue.

Jones et al. also found that a specific family of proteins, which helps to connect the collagen fibres, was increased in the tissue of patients with IPF. When these proteins were blocked with a newly developed drug, the collagen structure returned to normal and the stiffness of the tissue decreased. As a consequence, the lung capacity improved.

This suggests that treatment approaches that help to maintain a normal collagen structure, may in future prevent the stiffening of the lung tissue and so limit feed-forward mechanisms that drive progressive IPF. Moreover, it indicates that measurements of the structure of collagen rather than the its total concentration could serve as a more suitable indicator for the disease.
DOI: https://doi.org/10.7554/eLife.36354.002

## Results

### Increased mature collagen cross-linking but not total collagen content increases stiffness of IPF tissue

We performed an integrated biochemical and biomechanical characterisation comparing human IPF lung tissue with age-matched control lung tissue. We first used atomic force microscopy (AFM) canti-lever-based microindentation to assess lung tissue stiffness at the micrometre-scale. IPF tissue was significantly stiffer than control tissue (*Figure 1A*), with spatially heterogeneous changes in stiffness in IPF tissue including highly localised areas of increased stiffness which were not present in control lung tissue (*Figure 1B and C*), consistent with the known histopathological heterogeneity of IPF tissue (*Raghu et al., 2011*).

We then explored the relative contribution of collagen amount and/or post-translational modifications to these differences in stiffness. Whilst an increase in fibrillar collagen was suggested by second harmonic generation imaging of IPF lung tissue (*Figure 1D*), quantitation of total collagen concentration by hydroxyproline assay showed no difference in mean total collagen in IPF tissue relative to control tissue following normalisation to either dry weight or to total protein (*Figure 1E* and *Figure 1—figure supplement 1*); in addition, no dependence of lung tissue stiffness on collagen concentration was found (*Figure 1F*). In contrast, differences in the expression of collagen cross-linking enzymes were identified in IPF tissue. Whilst mRNA expression levels of LOX and LOXL1 were unchanged (*Figure 2A and B*), there were significant increases in the relative expression of LOXL2, LOXL3, and LOXL4 (*Figure 2C–E*). This was associated with detection of increased amine oxidase activity in IPF lung tissue sections (*Figure 2F*).

Consistent with previous reports of increased LH2 expression in fibrotic tissue (*Brinckmann et al., 1999*; *van der Slot et al., 2003*), we identified increased expression of LH2 in IPF tissue (*Figure 3A and B*) suggesting the potential for altered collagen cross-linking involving hydroxylysine residues. Therefore, we quantified immature (dihydroxylysinonorleucine (DHLNL) and hydroxylysinonorleucine (HLNL)) and mature trivalent (deoxypyridinoline (DPD) and pyridinoline (PYD)) hydroxyallysine-derived collagen cross-links. This revealed a significant increase in the density of immature divalent

(*Figure 3C*) and mature trivalent (*Figure 3D*) cross-links in IPF tissue, with an increase in the relative ratio of immature to mature cross-links in IPF tissue (*Figure 3E*) consistent with increased collagen metabolism in IPF tissue and an increase in the DHLNL to HLNL ratio (*Figure 3F*) consistent with increased lysyl hydroxylase activity. No dependence of lung tissue stiffness on immature cross-link density was found (*Figure 3G*), although a trend towards a correlation with DHLNL cross-link density alone was observed (correlation 0.41 p=0.09). In contrast, mature pyridinoline collagen cross-link density was strongly and significantly associated with increasing lung tissue stiffness (correlation 0.72 p=0.0007) (*Figure 3H*). Together, these results identify that in IPF lung tissue, differential expression of LH2 and LOXL-family members is associated with increased hydroxyallysine-derived pyridinoline collagen cross-links and this, rather than collagen concentration, is a primary determinant of increased tissue stiffness.

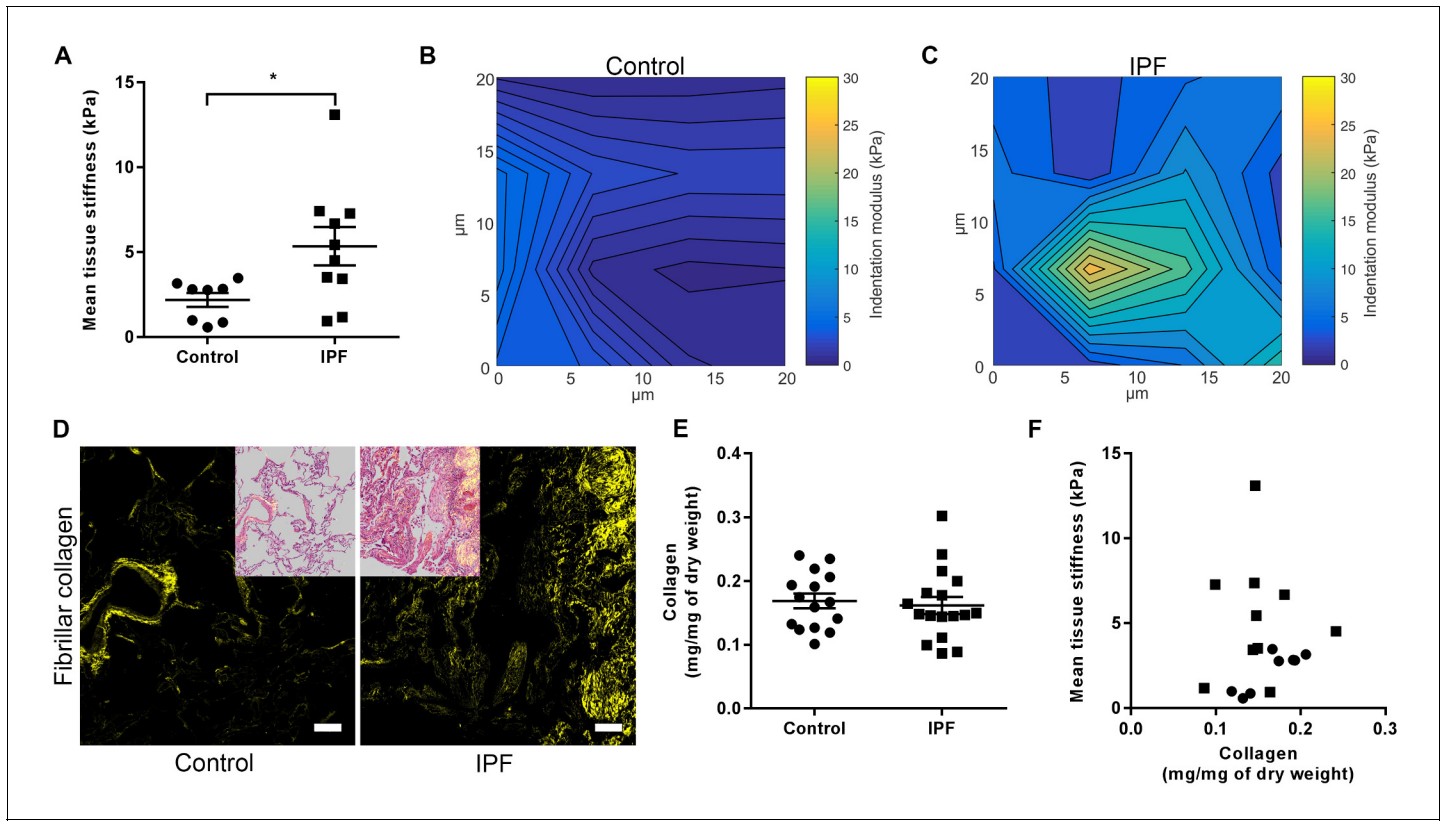

**Figure 1.** IPF lung tissue has increased stiffness which is not dependent on total collagen content. (A) Tissue stiffness of control (n = 8 donors, •) and IPF (n = 10 donors, ■) lung tissue determined by micro-indentation atomic force microscopy (AFM). Data presented as mean stiffness of each donor determined from force-displacement curve measurements (n = 80–150 measurements per donor). Bars are mean ± s.e.m. *p<0.05 by Student's *t*-test with Welch's correction for unequal variances (two-tailed). (B, C) Representative elastograph images from AFM cantilever-based microindentation analyses of control and IPF lung tissue, respectively. Axis labels indicate spatial scale in micrometres and colour bars the indentation modulus in kPa. The coefficient of variation in (B) is 66.6% and in (C) is 74.6%. (D) Representative second harmonic generation images (SHG) of control or IPF lung tissue. Fibrillar collagen is visualised in yellow. Inset image shows histological staining of an adjacent section using hematoxylin and eosin with the SHG image overlaid. Scale bar = 100 μm. (E) Total collagen content of control (n = 15) and IPF lung tissue (n = 17) normalised by dry weight. Bars are mean ±s.e.m. p=0.70 by Student's *t*-test (two-tailed). (F) Paired data from (A) and (E) were plotted to determine the dependency of mean tissue stiffness on collagen content (Spearman correlation coefficient: r = 0.16; p=0.52).

DOI: https://doi.org/10.7554/eLife.36354.003

The following figure supplement is available for figure 1:

**Figure supplement 1.** IPF lung tissue total collagen content.

DOI: https://doi.org/10.7554/eLife.36354.004

## Individual collagen fibrils from IPF tissue have altered structural and mechanical properties

Given the finding of increased collagen cross-linking in IPF, the morphological and biomechanical properties of individual collagen fibrils were analysed by AFM cantilever-based nanoindentation, following enzymatic extraction using an established methodology (*Andriotis et al., 2014*) (*Figure 4A and B*). Overall, IPF collagen fibrils were stiffer but exhibited a greater range of stiffness measurements compared with control lung collagen fibrils (*Figure 4C*). They also showed a skewed size distribution with a significantly smaller median diameter (*Figure 4D and E*) and had a lower fibril swelling ratio (hydrated to air) (*Figure 4F*). These data identify that in human lung fibrosis perturbation of collagen homeostasis occurs at the nanometre-scale, with individual collagen fibrils being structurally and functionally abnormal.

## LOXL2/LOXL3-selective inhibition of collagen cross-linking reduces tissue stiffness

To explore the underlying mechanisms of altered fibrillogenesis and collagen structure-function in IPF, we first established a novel long-term 3D in vitro model of lung fibrosis using primary human lung fibroblasts treated with the pro-fibrotic cytokine TGF-$\beta_1$. We studied lung fibroblasts from patients with IPF given previous studies identifying aberrations in their fibrogenic responses (*Ramos et al., 2001*). An advantage of this model is that it allows not only direct evaluation of cross-

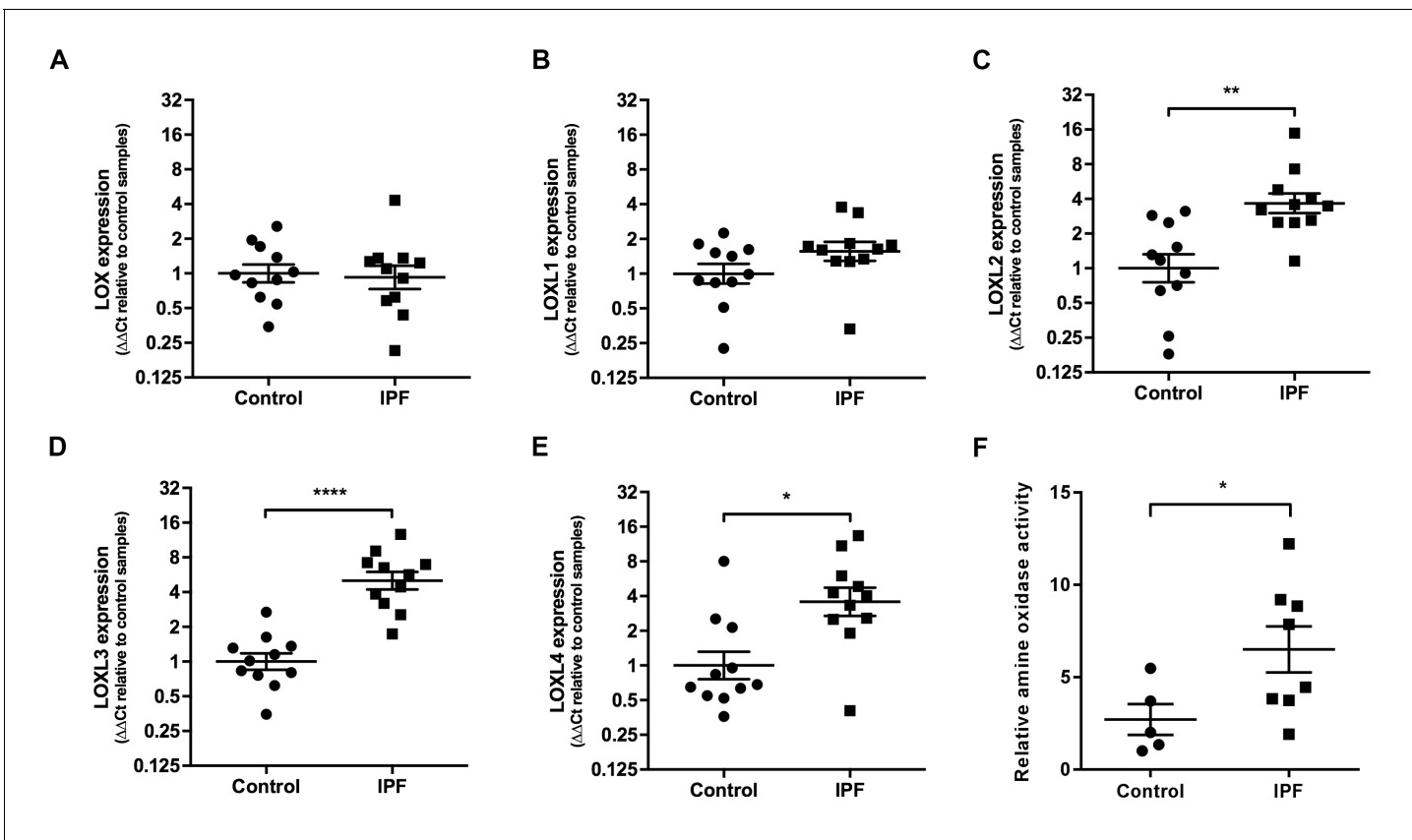

**Figure 2.** Lysyl oxidase-like (LOXL) enzymes and amine oxidase activity are increased in IPF tissue. (A–E) Expression of *LOX, LOXL1, LOXL2, LOXL3* and *LOXL4* was analysed in control (•) and IPF (■) lung tissue (n = 11 individual donors per group) using the ΔΔCt method. Bars indicate geometric means. LOX p=0.79; LOXL1 p=0.22 *p<0.05; **p<0.01; ***p<0.001; ****p<0.0001 by multiple *t*-test of log transformed data using the Holm-Sidak method to adjust for multiple comparisons (two-tailed). SD was not assumed to be consistent between groups. (F) Amine oxidase activity as measured in situ from control (n = 5 individual donors) or IPF lung tissue (n = 8 individual donors). Semi-quantitative analysis was performed using Fiji and normalised to one control sample. Bars are mean ±s.e.m. *p<0.05 by Student's *t*-test with Welch's correction for unequal variances (2-tailed).
DOI: https://doi.org/10.7554/eLife.36354.005

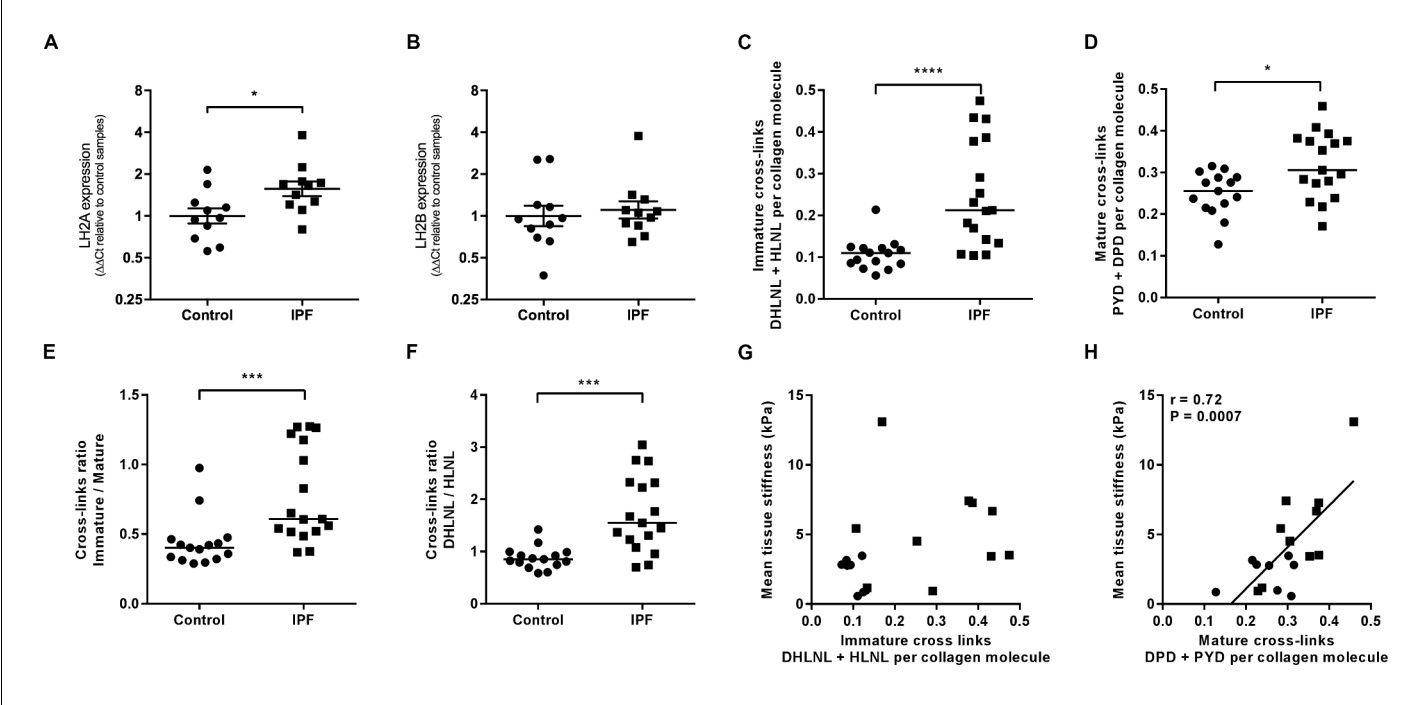

**Figure 3.** Hydroxyallysine-derived collagen cross-link density is increased in IPF lung and is correlated with tissue stiffness. (**A, B**) Lysyl hydroxylase 2 (*LH2A* and *LH2B*) expression was analysed in control (•) and IPF (■) lung tissue (n = 11 donors per group) using the ΔΔCt method. Bars indicate geometric means. *p<0.05 by multiple *t*-test of log transformed data using the Holm-Sidak method to adjust for multiple comparisons (two-tailed). SD was not assumed to be consistent between groups. (**C–F**) Hydroxyallysine-derived collagen cross-link analysis of IPF (n = 17 donors) and control (n = 15 donors) lung tissue, normalised to total collagen content, as determined by UHPLC-ESI-MS/MS. Each data point represents an individual donor. (**C**) Total immature divalent cross-links (DHLNL + HLNL); (**D**) Total mature trivalent cross-links (PYD + DPD); (**E**) Ratio of immature to mature cross links; (**F**) Ratio of DHLNL to HLNL immature divalent cross-links. Bars indicate median values. *p<0.05; **p<0.01; ***p<0.001; ****p<0.0001 by Mann-Whitney *t*-test (two-tailed). (**G, H**) Scatterplots of paired data from *Figure 1A* and *Figure 3C* or *Figure 3D* showing dependency of mean tissue stiffness on total immature (r = 0.364, p=0.137) or total mature (r = 0.723, p=0.0007) collagen cross-link density, respectively (Multivariate correlation analysis including mean tissue stiffness, normalised collagen and immature and mature cross-links).
DOI: https://doi.org/10.7554/eLife.36354.006

linking and its inhibition but also direct measurement of their influence upon tissue biomechanics. Over 6 weeks of culture, the model shows a progressive increase in collagen content and mature collagen cross-links in association with expression of all LOX/LOXL family members (*Figure 5A–D*); histochemical staining of the resultant tissue construct showed fibrillar collagen deposition (*Figure 5E*) similar to that seen in fibroblastic foci, the site of active fibrogenesis in IPF (*Figure 5F*) (*Jones et al., 2016*).

Given our finding of differential LOXL enzyme expression, we evaluated the contribution of these enzymes to collagen cross-linking and tissue stiffness using haloallylamine based compounds which have previously been demonstrated to be amine oxidase inhibitors that show selectivity towards individual enzymes (*Schilter et al., 2015*; *Chang et al., 2017*). We tested a number of these compounds for inhibition of each LOX/LOXL family member, and identified that one of these inhibitors, PXS-S2A, which was previously reported as a LOXL2-selective inhibitor (*Chang et al., 2017*), is a dual LOXL2 and LOXL3-selective inhibitor. PXS-S2A irreversibly inhibits LOXL2 and LOXL3 ($IC_{50}$ 0.005 μM and 0.016 μM respectively) but is reversible and more than a hundred times less potent for inhibition of LOX and LOXL1 (*Figure 6A*). Using the inhibitor in the in vitro model, we observed a dose-dependent inhibition of collagen cross-links (*Figure 6B–D*), with a greater than 50% reduction in mature pyridinoline cross-links (*Figure 6D*) with 0.1 μM PXS-S2A, a concentration which completely inhibits LOXL2 and LOXL3 but has minimal effects on LOX and LOXL1 (*Figure 6A*). At a pan LOX/LOXL inhibitory concentration of 10 μM, the reduction in cross-links was comparable to that observed with β-aminoproprionitrile (BAPN), a non-selective LOX/LOXL inhibitor

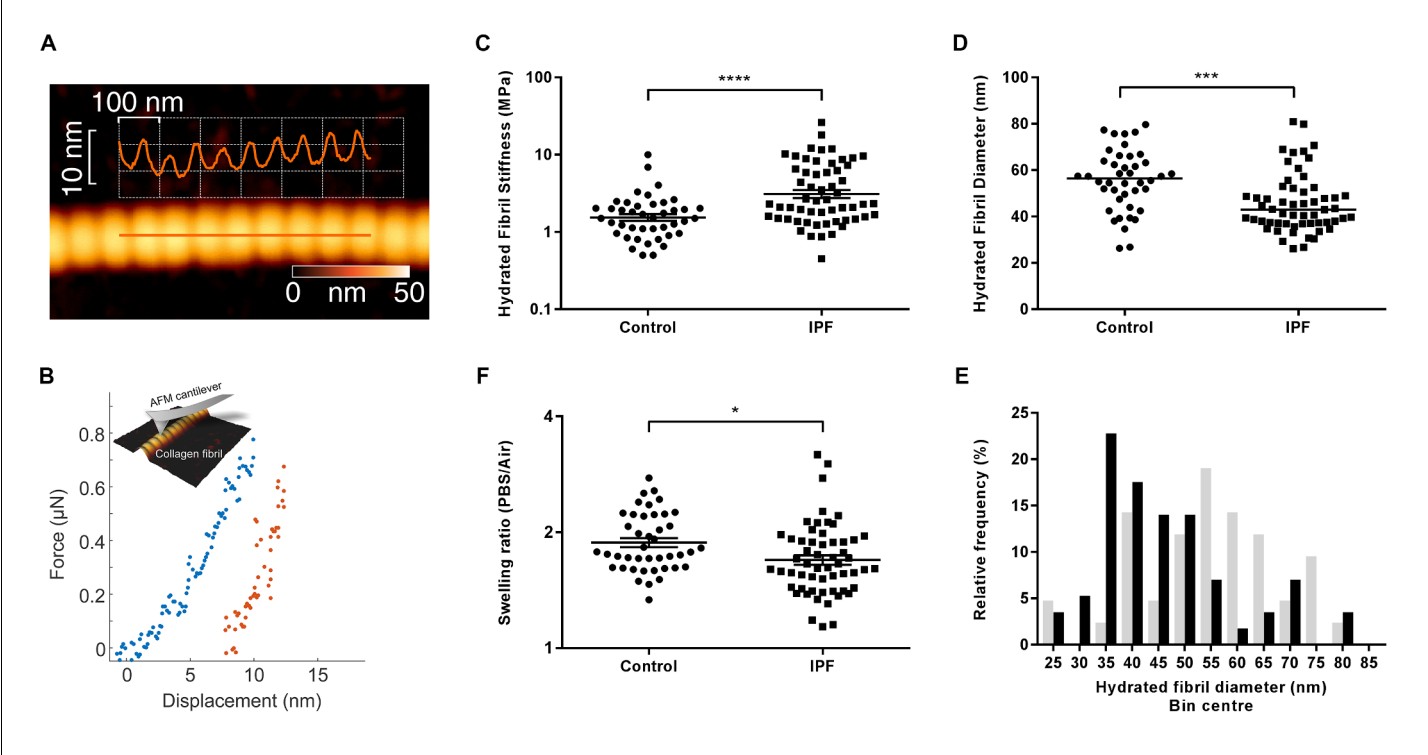

**Figure 4.** Collagen fibrils from IPF tissue have altered fibril diameter and stiffness. (A) Atomic force microscopy (AFM) height topography image of a collagen fibril and corresponding height topography long axis profile showing characteristic D-periodicity (~67 nm). (B) Force displacement curve (blue: loading, red: unloading) of an individual fibril tested using AFM cantilever-based nanoindentation. (C) Indentation modulus of collagen fibrils (3–7 fibrils per donor) from control (n = 42 fibrils from eight donors) or IPF lung tissue (n = 57 fibrils from 10 donors) under hydrated conditions; each data point represents the mean of 30 to 50 force-displacement curves per fibril. Bars are mean ±s.e.m of log transformed data. ****p<0.0001 by Student's *t*-test with Welch's correction for unequal variances of log transformed data (two-tailed). (D) Fibril diameter as determined by measurement of fibril height under hydrated conditions. Bars are median; ***p<0.001 by Mann-Whitney *t*-test (two-tailed). (E) Frequency distribution of hydrated collagen fibril diameters measured for control (light grey bars) and IPF lung tissue (black bars). (F) Fibril swelling as determined by the ratio of the diameters under hydrated and dry conditions. Bars are mean ±s.e.m of log transformed data. *p<0.05 by Student's *t*-test of log transformed data (two-tailed).

DOI: https://doi.org/10.7554/eLife.36354.007

(*Trackman, 2016*). No reduction in total collagen content was observed following treatment with PXS-S2A or BAPN (*Figure 6E*).

The biomechanical consequence of inhibiting collagen cross-linking was then investigated with parallel plate compression testing. The LOXL2/LOXL3 inhibitor dose-dependently reduced tissue stiffness as measured by Young's modulus, achieving maximal inhibition of stiffness as compared with BAPN at a pan LOX/LOXL inhibitory concentration of 10 μM. However, almost 70% of maximal reduction in stiffness (as compared with BAPN) was already achieved at 0.1 μM PXS-S2A which is selective for LOXL2/LOXL3 (*Figure 6F and G*, *Figure 6—figure supplement 1*, and *Video 1*), suggesting that inhibition of LOXL2/LOXL3 is sufficient to achieve a substantial reduction in tissue stiffness. Consistent with the IPF tissue findings, stiffness showed no dependency on collagen content (*Figure 6H*) whilst it correlated positively with collagen cross-link density including both immature (*Figure 6I*) and mature pyridinoline cross-links (*Figure 6J*). Together, these data identify pyridinoline cross-link density to be a significant determinant of tissue stiffness and identify LOXL2/LOXL3 enzyme activities as essential contributors to this process.

## Selective LOXL2/LOXL3 inhibition normalises collagen fibril assembly

We next assessed the impact of selective LOXL2/LOXL3 inhibition upon collagen morphology in the in vitro fibrosis model. When visualised by polarised light Picrosirius red microscopy, highly organised collagen fibrils were evident in vehicle-treated fibrotic control cultures as well as in those

**Figure 5.** In vitro modelling of fibrillar collagen production and cross-linking in IPF. We utilised a long-term 3D in vitro model of lung fibrosis using primary human lung fibroblasts from patients with IPF treated with the pro-fibrotic cytokine TGF-$\beta_1$ cultured for up to 6 weeks. (A–C) Characterisation of (A) total collagen normalised to total protein, (B) immature divalent (DHLNL) and (C) mature trivalent (PYD and DPD) hydroxyallysine-derived collagen cross-links at the culture times indicated following addition of TGF-$\beta_1$. Bars are mean +range (n = 2 IPF donors). (D) Relative gene expression analysis of *LOX*, *LOXL1*, *LOXL2*, *LOXL3* and *LOXL4* using the ΔΔCt method (n = 3 IPF donors, two experiments per donor). (E, F) Masson's trichrome stain of histological sections from (E) the in vitro model at 6 weeks and (F) IPF lung tissue including a fibroblastic focus. Blue staining identifies fibrillar collagen. Scale bars are 100 µm (main image) and 500 µm (inset, showing location of enlarged image).

DOI: https://doi.org/10.7554/eLife.36354.008

treated with 0.1 µM PXS-S2A; this contrasted with the marked disorganisation observed with BAPN (*Figure 7A*). Based on these findings, we performed ultrastructural analysis of the collagen fibrils using electron microscopy (*Figure 7B*). At 0.1 µM PXS-S2A, a 12.3% mean increase in fibril diameter was observed (*Figure 7C*), comparable with the 18% difference in fibril size observed in our AFM analyses comparing control and IPF collagen fibrils. Increasing the concentration of PXS-S2A to 0.5 µM caused a further small increase in fibril diameter; however, with complete LOX/LOXL inhibition using BAPN there was a broadening of fibril diameter distribution and a marked dysregulation of fibril structure including irregular profiles (*Figure 7B and C*). In keeping with this, BAPN markedly increased collagen solubility, with only 14% of the total collagen remaining

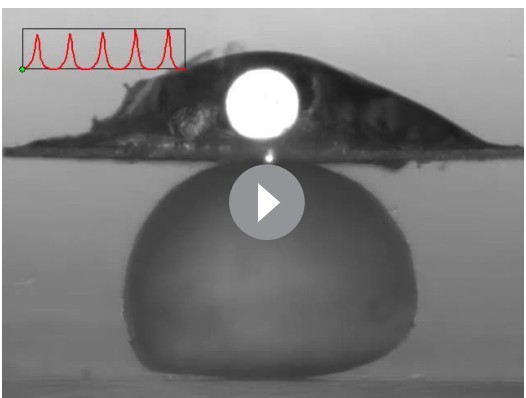

**Video 1.** 10x Time-lapse movie of an in vitro model fibrosis sample during testing by parallel plate compression.

DOI: https://doi.org/10.7554/eLife.36354.011

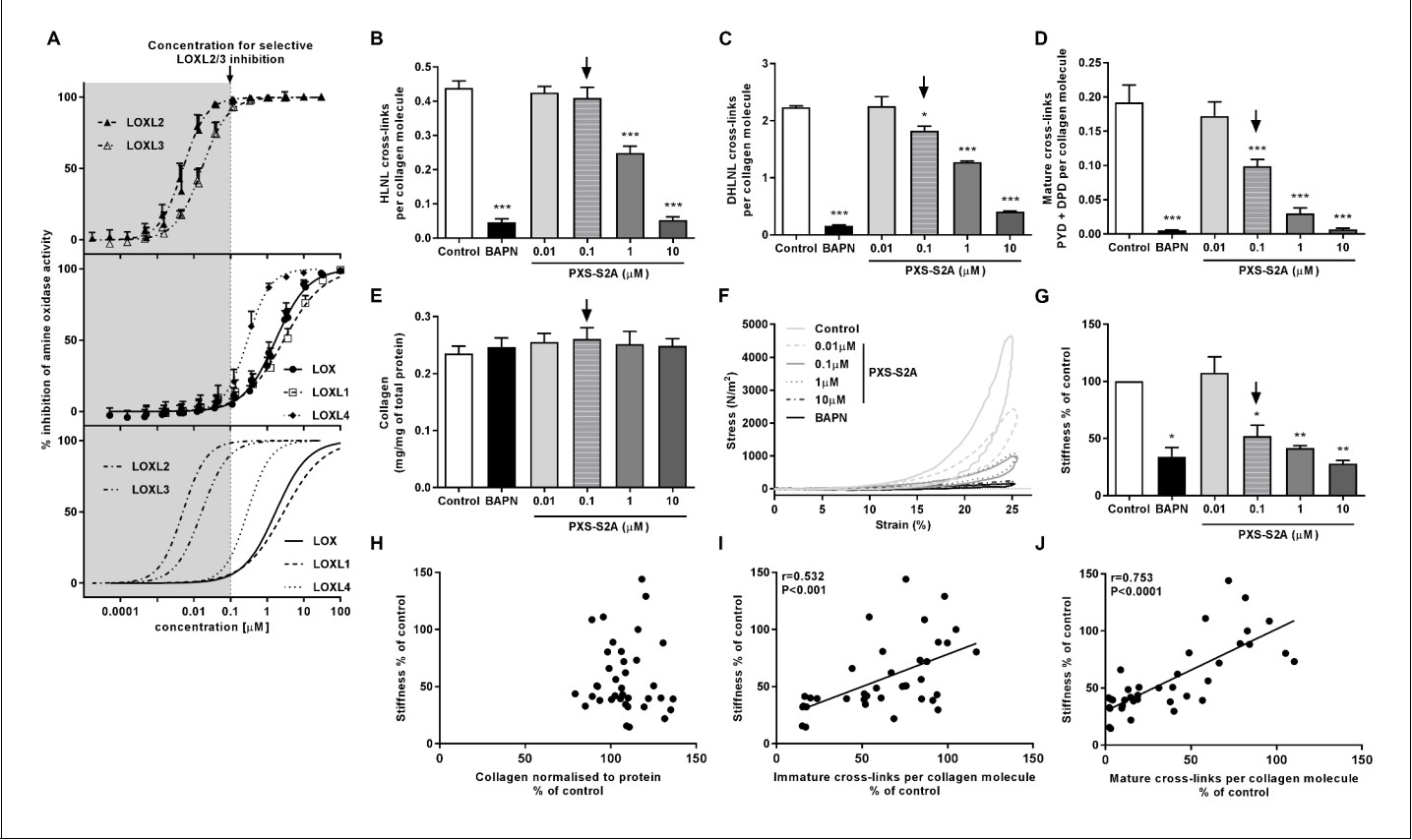

**Figure 6.** Selective inhibition of LOXL2/LOXL3 reduces hydroxyallysine-derived collagen cross-links and tissue stiffness. (**A**) PXS-S2A inhibition curves against LOXL2 and LOXL3 (upper panel), LOX, LOXL1, LOXL4 (middle panel), and all LOX/LOXL enzymes (lower panel). Downward arrows in (**A–E, G**) identify LOXL2/LOXL3-selective dose. (**B–G**) Lung fibroblasts from IPF patients (n = 3 donors, two experiments per donor) were used in the 3D model of fibrosis in the presence of 1 mM BAPN, PXS-S2A (at concentrations indicated) or vehicle control for 6 weeks. Bars are mean +s.e.m. (**B–D**) Hydroxyallysine-derived collagen cross-link analysis: (**B**) HLNL determined by LC-MS; (**C**) DHLNL determined by LC-MS; (**D**) Total mature (PYD +DPD) cross-links determined by ELISA. *p<0.05; **p<0.01; ***p<0.001 by one-way repeated measures ANOVA with Dunnett's post-test. (**E**) Total collagen content determined by hydroxyproline assay. (**F–G**) Parallel plate compression testing of the 3D culture of lung fibrosis. (**F**) Typical stress versus strain plots following treatment with LOXL2/LOXL3 inhibitor or BAPN. (**G**) Young's modulus. *p<0.05; **p<0.01 by one sample t-test. (**H–J**) Scatterplots showing dependency of tissue stiffness on (**H**) collagen content (r = −0.0272; p=0.873), (**I**) immature (r = 0.532; p=0.0007) or (**J**) mature (r = 0.753; p<0.0001) collagen cross-link density following PXS-S2A treatment (Multivariate correlation including mean tissue stiffness, normalised collagen and immature and mature cross-links).

DOI: https://doi.org/10.7554/eLife.36354.009

The following figure supplement is available for figure 6:

**Figure supplement 1.** Parallel plate compression testing and analysis of the in vitro fibrosis model: Example for a single experiment.
DOI: https://doi.org/10.7554/eLife.36354.010

insoluble after proteolytic digestion, whilst with 0.1 µM PXS-S2A only a small increase in collagen solubility was observed suggesting that inhibition of LOXL2/LOXL3 did not significantly disrupt fibril stability (*Figure 7D*). Together, these results identify that LOX/LOXL activity is an essential regulator of collagen fibril assembly and that selective LOXL2/LOXL3 inhibition is sufficient to significantly reduce tissue stiffness and normalise collagen fibril architecture.

## Small molecule LOXL2/LOXL3-selective inhibition as a treatment for lung fibrosis

To extend our in vitro studies, the efficacy of an orally bioavailable small molecule inhibitor (PXS-S3B, IC$_{50}$0.008 µM and 0.020 µM for LOXL2 and LOXL3, compared with 0.118 µM, 1.130 µM and 1.710 µM for LOXL4, LOX and LOXL1 respectively) was then assessed at LOXL2/LOXL3-selective doses in an experimental animal model of lung fibrosis. Control experiments using the 3D in vitro

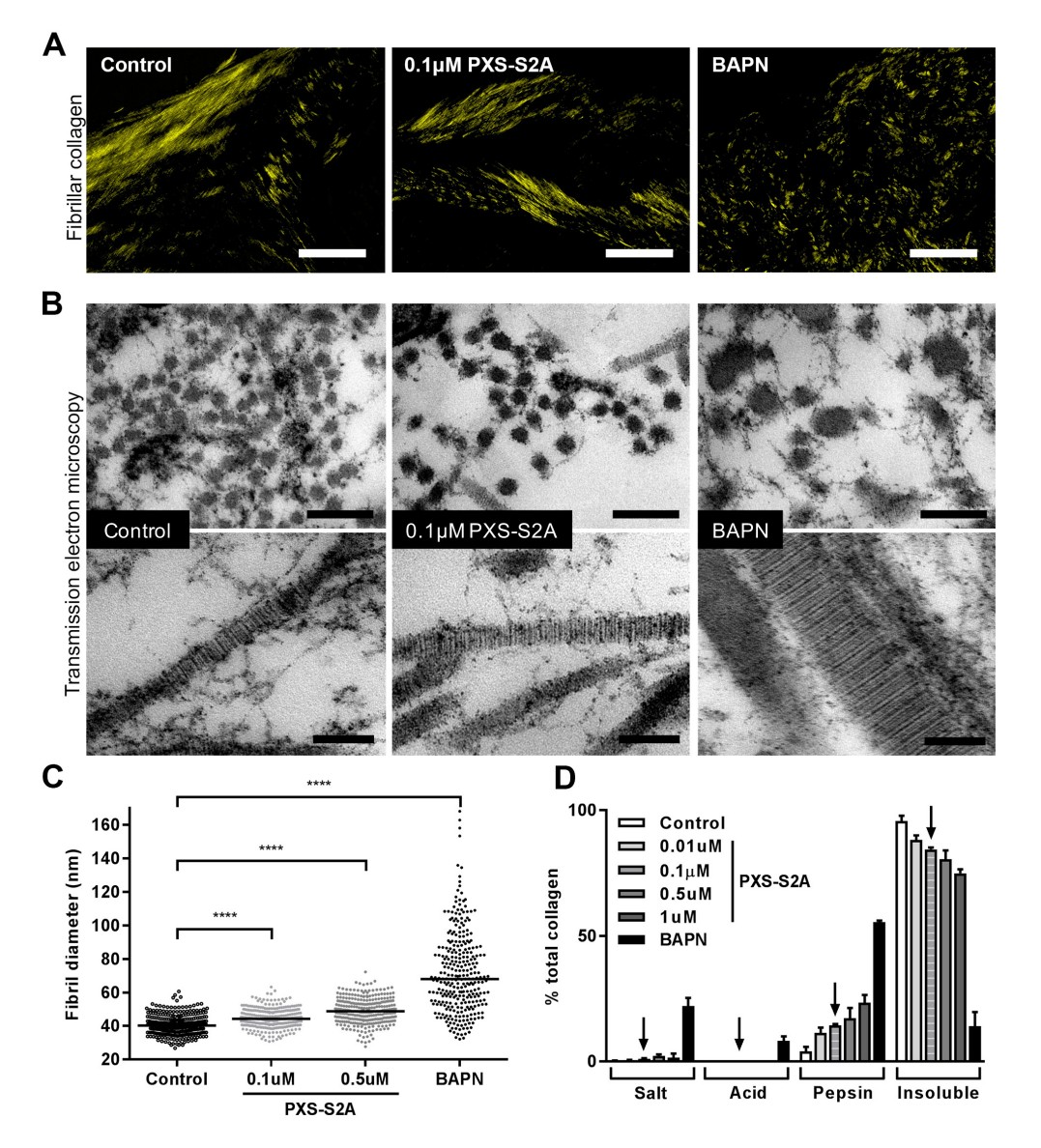

**Figure 7.** Selective LOXL2/LOXL3 inhibition modifies collagen fibril assembly. IPF fibroblasts were grown for 6 weeks in the 3D in vitro model of fibrosis with vehicle control, concentrations of PXS-S2A or 1 mM BAPN, as indicated. (**A**) Representative images of histological sections stained with picrosirius red and viewed under plane polarised light, scale bar: 50 μm. (**B**) Representative transmission electron microscopy images of collagen fibrils in transverse (upper panel, scale bar 200 nm) and longitudinal (lower panel, scale bar 100 nm) cross-section. Images are representative of the mean fibril diameter. (**C**) Collagen fibril diameter measured in transverse section (300 fibrils per treatment measured by a blinded observer from one experiment). Data are representative of measurements from two independent experiments. Bars show the median value. ****$p < 0.0001$ by non-parametric one-way ANOVA (Kruskal-Wallis test) with Dunn's multiple comparison test. (**D**) Collagen solubility of whole cultures assessed by sequential extractions using TBS, 0.5M acetic acid, and pepsin. Each fraction and the final insoluble residue were hydrolysed and assayed for hydroxyproline. Bars are mean +range of two IPF donors.

DOI: https://doi.org/10.7554/eLife.36354.012

model of fibrosis confirmed the effect of PXS-S3B on collagen cross-linking and tissue stiffness was comparable to PXS-S2A (*Figure 8—figure supplement 1*).

Given the importance of TGF-β as a driver of fibrosis, we used an in vivo model of lung fibrosis driven by transient overexpression of active TGF-$\beta_1$ by adenoviral vector gene transfer, which results in severe progressive fibrosis and recapitulates several features of pulmonary fibrosis in human disease (*Sime et al., 1997*). Rats received either a replication-defective adenoviral vector producing

active TGF-$\beta_1$ (AdTGF-$\beta_1$) or vector only control (AdDL) on day 0, and then oral administration of PXS-S3B (15 or 30 mg/kg/day) from day 1 to day 28 at which point lung function was assessed before the animals were euthanised for biochemical and histological analysis of the lungs. There was no effect of PXS-S3B on total collagen content of the lungs (*Figure 8A*), but there was a significant reduction in fibrosis compared to vehicle-treated AdTGF-$\beta_1$ rats, as assessed by modified Ashcroft scores (*Figure 8B and C*), and a significant improvement in lung function (i.e. a reduction in lung stiffness as measured by pressure-driven pressure volume-loops and elastance) at day 28 (*Figure 8D and E*), reflecting our in vitro finding that LOXL2/LOXL3 inhibition reduces tissue stiffness. This was accompanied by a reduction in mature pyridinoline collagen cross-links compared with the AdTGF-$\beta$-treated rats (*Figure 8F and G*), as well as immature (DHLNL + HLNL) cross-links (*Figure 8H*). We also determined that the DHLNL:HLNL ratio was reduced in the treated rats suggesting a reduction in lysine hydroxylation (*Figure 8I*). Consistent with this, analysis of mRNA expression in the lungs of animals treated with LOXL2/LOXL3 inhibitor, revealed a decrease in LH2, which is required for telopeptide hydroxyallysine-derived cross-links. In addition, expression of other modulators of fibrillogenesis including collagen V, TGM2, dermatopontin, fibulin, fibrillin, and periostin (*Figure 8J*) were reduced. Thus, inhibition of LOXL2/LOXL3 not only modifies collagen cross-linking but also modulates fibrillar collagen homeostasis. No adverse effects of treatment with the LOXL2/LOXL3 inhibitor were evident in the model.

## Discussion

One of the critical issues in the pathobiology of fibrosis is an insufficient understanding of the structure-function relationship of the ECM in human disease. A major goal of this study was to perform an integrated biomechanical and biochemical characterisation of fibrillar collagen and its pathological role in human fibrosis. We demonstrate that dysregulated collagen fibrillogenesis is a key pathway of human lung fibrosis at the time of diagnosis, with structurally and functionally abnormal collagen – rather than collagen content - increasing lung tissue stiffness as a consequence of pathologic pyridinoline cross-linking. To advance our mechanistic understanding of these findings, we employed small molecule LOXL inhibitors and demonstrate that dual inhibition of LOXL2 and LOXL3 is sufficient to prevent accumulation of these pyridinoline cross-links. This normalises the structure and biomechanical properties of fibrillar collagen, and reduces tissue stiffness without the need to significantly disrupt fibril assembly. Our findings suggest that this targeted inhibition reduces the disruption of mechano-homeostasis to limit the self-sustaining effects of ECM on progressive fibrosis.

A thought provoking finding of our studies is that total collagen concentration, as determined by measurement of hydroxyproline that quantifies all collagen irrespective of whether it is assembled into fibrils, was not significantly increased in IPF tissue. Whilst Selman *et al* identified an increase in collagen of fibrotic lung biopsy samples (*Selman et al., 1986*), other studies have reported no significant increase in collagen deposition (*Nkyimbeng et al., 2013*; *Fulmer et al., 1980*; *Westergren-Thorsson et al., 2017*), or have reported increased collagen deposition only in late stage disease at autopsy, but not in diagnostic lung biopsy samples (*Kirk et al., 1986*). This suggests that, at the time of diagnosis, any increase in collagen is paralleled by changes in other proteins (most likely other ECM and myofibroblastic cellular proteins), but that by late-stage fibrosis there may be a progressive increase in collagen accumulation, perhaps as the fibrotic tissue becomes more acellular and/or the collagen resists degradation. Furthermore, although we identified an increase in stiffness of IPF lung tissue using AFM micro-indentation, this was not related to collagen concentration. Rather, increased mature fibrillar collagen pyridinoline cross-linking secondary to dysregulated post-translational modification was a primary determinant of tissue stiffness.

Fibrillar collagen is a predominant component of the ECM which provides tissues with essential tensile strength by forming a hierarchical assembly of substructures dependent upon their cross-linking: tropocollagen molecules assemble into fibrils and the fibrils further assemble into fibres. Here, we identified that collagen hierarchical assembly is altered at the nanoscale/ultrastructural level in IPF. Collagen fibrils were structurally and functionally abnormal with reduced diameter and increased stiffness, suggesting that at a molecular level they have the potential to influence the microenvironment of individual cells to affect their biomechanical sensing. Covalent cross-links have been proposed to increase fibril stiffness by forming a lateral network of linkages (*Bailey, 2001*; *Miles et al., 2005*). In models of artificially cross-linked collagen fibrils, it was observed that by drawing the

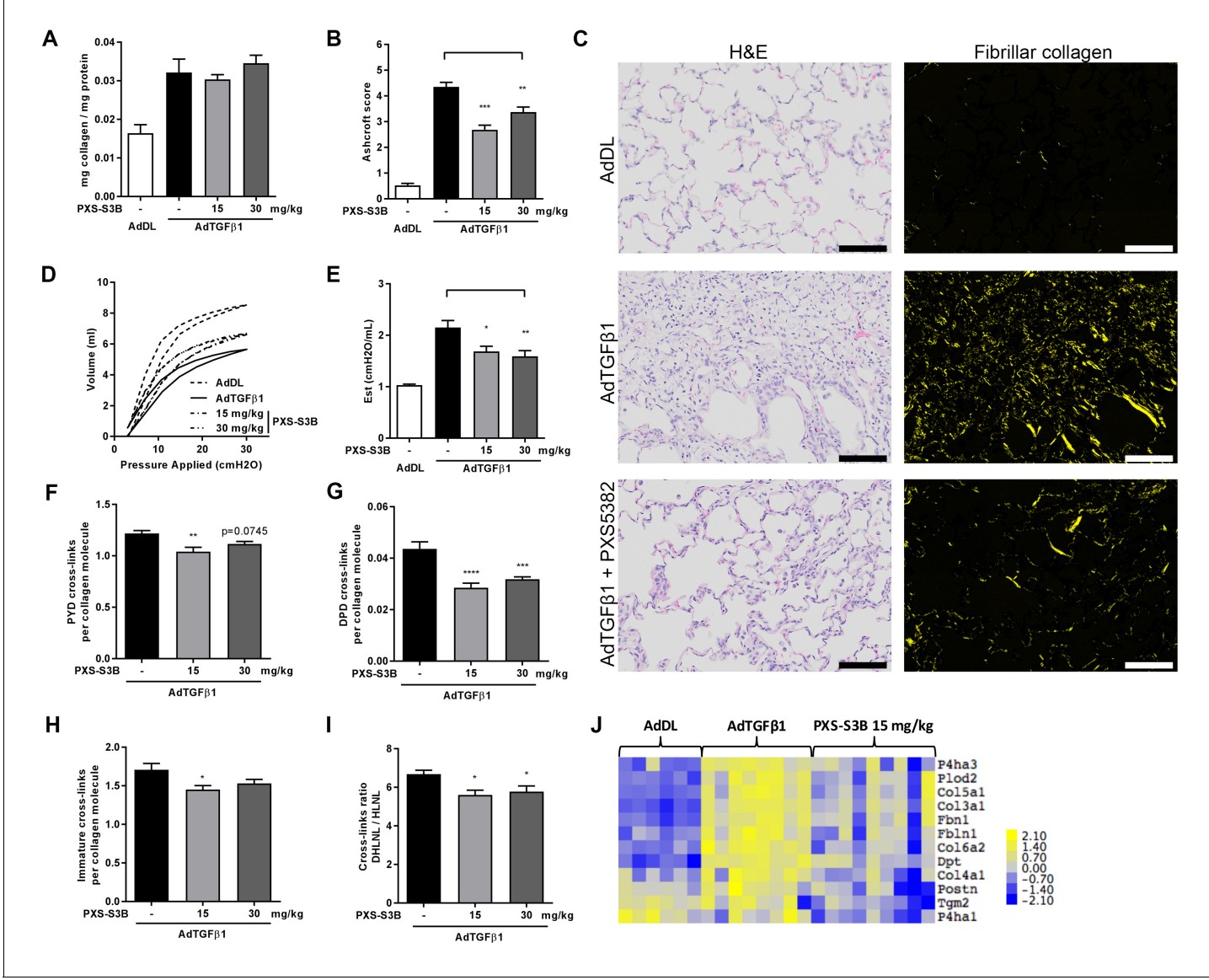

**Figure 8.** Selective LOXL2/LOXL3 inhibition reduces fibrosis and improves lung function in an in vivo model of lung fibrosis driven by TGF-β. Rats received either empty vector control (AdDL) (n = 6) on Day 0 or adenoviral vector producing active TGF-β₁ (AdTGF-β1) to induce progressive pulmonary fibrosis. AdTGF-β1 groups received either vehicle (PBS, n = 8) or PXS-S3B (15 mg/kg/day or 30 mg/kg/day, n = 9 per group) dosed daily orally from day 1 to day 28 when animals were sacrificed. (A) Total collagen content normalised to total protein content. (B) Fibrotic changes in left lung quantified by modified Ashcroft score. (C) Representative histological sections stained with H and E (left hand panel) or picrosirius red viewed under plane polarised light (fibrillar collagen, right hand panel) (Scale bar = 100 μm). (D–E) Day 28 lung function prior to sacrifice: (D) Representative lung pressure-volume loops. (E) Lung elastance. (F–I) Hydroxyallysine-derived collagen cross-link analysis determined by LC-MS: (F) PYD cross-links, (G) DPD cross-links, (H) immature divalent (DHLNL +HLNL) cross-links; (I) ratio of DHLNL to HLNL cross-links; Bars are mean +s.e.m. *p<0.05; **p<0.01; ***p<0.001; ****p<0.0001 by one-way ANOVA with Dunnett's multiple comparison test. (J) Transcriptional changes in fibrillogenesis genes in lung tissue shown in a heat map produced by treatment group supervised clustering.

DOI: https://doi.org/10.7554/eLife.36354.013

The following figure supplement is available for figure 8:

**Figure supplement 1.** Selective inhibition of LOXL2/LOXL3 with PXS-S3B reduces hydroxyallysine-derived collagen cross-links and tissue stiffness.

DOI: https://doi.org/10.7554/eLife.36354.014

collagen molecules closer together there was a reduction in water between molecules; as a consequence, the density of collagen fibrils increased and this was proposed to affect their mechanical properties (*Bailey, 2001*; *Miles et al., 2005*). Whether a similar effect occurs after enzymatic cross-

linking, especially pathological cross-linking involving pyridinoline cross-links, remains to be determined. However, the reduction in fibril diameter and lesser fibril swelling with hydration of IPF fibrils would be consistent with this concept. Whilst direct experimental evidence is required to determine whether collagen swelling is directly related to cross-linking, collagen hydration alone has been shown to significantly influence individual collagen fibril mechanics and the mechanism of molecular slippage and stretching within fibrils (*Andriotis et al., 2018*; *Gautieri et al., 2011*; *Yang et al., 2008*).

Several different mechanisms have been proposed to contribute to alterations in tissue stiffness including LOX/LOXL and transglutaminase 2 (TGM2)-mediated enzymatic cross-linking and non-enzymatic glycation. Here we have focussed on the LOX and LOXL enzymes which play key roles in the process of fibrillar collagen production, with expression being tightly controlled in normal development (*Trackman, 2016*). Whilst LOXL2 has been proposed to have pathologic roles in cancer and fibrosis (Barker et al.; *Barry-Hamilton et al., 2010*), and increased LOXL2 expression in IPF stroma has been observed previously, less is understood regarding LOXL3. Previous studies have identified that it is expressed in human IPF lung tissue, has amine oxidase activity for collagens, is induced by fibroblast culture on IPF cell-derived ECM, and is an enhancer and key regulator of integrin signalling in myofibre formation (*Kraft-Sheleg et al., 2016*; *Lee and Kim, 2006*; *Philp et al., 2018*). Recently, both LOXL2 and LOXL3 were identified to be crucial for fibroblast-to-myofibroblast transition in in vitro models of lung fibrosis (*Aumiller et al., 2017*). In keeping with our finding of increased LOXL expression together with increased LH2 expression in IPF tissue, we identified an increase in mature hydroxyallysine-derived cross-links in IPF lung tissue, consistent with a previous report of an increase in mature DPD cross-links in fibrotic lung tissue (*Last et al., 1990*). The extent of lysine hydroxylation can vary from 15 to 90% depending on the collagen types and, even within type I collagen, it varies significantly from tissue to tissue and under physiological or pathological conditions (*Yamauchi and Sricholpech, 2012*). Although it is unknown whether LOXL2 and/or LOXL3 have a substrate preference for hydroxylysine, the shift from a skin (allysine-derived) to a bone-type (hydroxyallysine-derived) of collagen cross-link appears of pathogenetic importance in IPF.

Various approaches have investigated the roles of collagen cross-linking by LOX/LOXL enzymes in fibrosis. Multiple studies have tested BAPN as a broad-spectrum LOX/LOXL inhibitor (*Trackman, 2016*), although this approach has no therapeutic potential as it does not enable specific enzyme targeting and has undesirable effects on LOX leading to unwanted vascular and skeletal side-effects (*Wawzonek et al., 1955*). A monoclonal antibody which binds and partially inhibits LOXL2, showed efficacy in the bleomycin mouse model of lung fibrosis (*Barry-Hamilton et al., 2010*). However, recently, a Phase 2 study in IPF of simtuzumab, the humanised version of this antibody, was terminated due to lack of efficacy (*Raghu et al., 2017*). Our finding of differential expression of LOXL2, LOXL3 and LOXL4 in IPF tissue suggests that therapeutic targeting of additional LOXL family members is required to inhibit pathologic collagen cross-linking. To investigate this finding, we used dual LOXL2 and LOXL3-selective inhibitors which irreversibly inhibit LOXL2 and LOXL3 but are reversible and more than a hundred times less potent for inhibition of LOX and LOXL1. Characterisation in a novel 3D in vitro model of human fibrosis enabled direct measurement of the influence of LOXL2/LOXL3 inhibition upon pathological changes in collagen structure-function.

A key finding of our study is that approximately 70% of maximal reduction in tissue stiffness in vitro occurred following selective inhibition of only LOXL2/LOXL3 when pyridinoline cross-links were significantly reduced and fibril diameter was normalised, whilst fibril architecture was preserved and only a small increase in protease-sensitive collagen solubility was evident. Together our data suggest that in lung fibrosis reducing pathological collagen cross-linking normalises fibril architecture and is sufficient to reduce tissue stiffness without the need to completely disrupt fibril assembly. While LOXL4 was also increased in IPF, our dose-response findings from the parallel plate compression system suggest that it does not make a significant contribution to tissue stiffness. In comparison with selective LOXL2/LOXL3 inhibition in the current study, previous studies treating tendon constructs from days 14 to 21 with the pan LOX/LOXL inhibitor BAPN showed that collagen fibrils were irregular with widely dispersed diameters (*Herchenhan et al., 2015*) and we observed a similar effect of BAPN in the current study. Thus, while pan LOX/LOXL inhibition disrupts normal collagen fibrillogenesis, selective LOXL2/LOXL3 inhibition was sufficient to prevent pathological collagen crosslinking and restore fibrillar collagen homeostasis.

Although reduction in hydroxyproline measurements of collagen is recommended as a primary measure of anti-fibrotic efficacy during preclinical animal model testing of putative therapeutics for lung fibrosis (*ATS Assembly on Respiratory Cell and Molecular Biology et al., 2017*), our data from human tissue obtained at time of diagnosis identifies that effects on collagen fibrillogenesis are more relevant pathogenic mechanisms with abnormal collagen cross-linking and increased tissue stiffness likely to precede any accumulation of collagen in late-stage human pulmonary fibrosis. Consistent with our proposal, in a rat carbon tetrachloride-induced model of liver injury, increased liver stiffness was associated with increased collagen cross-linking and this was proposed to prime the liver to respond quickly to injury via mechanical feed-forward mechanisms (*Perepelyuk et al., 2013*; *Georges et al., 2007*).

The relevance of LOXL2/LOXL3-selective inhibition was demonstrated in a TGF-β-driven rat model of lung fibrosis. The model leads to persistent and severe interstitial fibrosis and does not have the confounding robust inflammatory response observed in the commonly studied bleomycin induced mouse model of fibrosis so enabling direct assessment of fibrotic responses (*Sime et al., 1997*; *ATS Assembly on Respiratory Cell and Molecular Biology et al., 2017*). Selective targeting of LOXL2 and LOXL3 in this model reduced pyridinoline collagen cross-links, reduced fibrosis, and improved lung function without a significant effect on hydroxyproline content. Whilst this did not completely restore lung function back to normal, there was a significant beneficial effect even in this model of fibrosis which is very strongly driven by TGF-β. An interesting finding was that inhibition of LOXL2 and LOXL3 reduced expression of genes known to influence collagen fibrillogenesis including LH2, collagen V, TGM2, dermatopontin, fibulin, fibrillin, and periostin. As these genes, as well as LOXL2 and LH2 are induced by TGF-β, one possible explanation is that TGF-β plays a causal role in fibrosis by disrupting collagen homeostasis to drive abnormal collagen cross-linking and promoting assembly of stiffer collagen fibrils. This change in stiffness initiates self-sustaining mechanosensory signalling pathways in a feed-forward mechanism to drive fibroblast activation and progressive fibrosis (*Herrera et al., 2018*; *Tschumperlin et al., 2018*). Given the important contribution of fibrillar collagen in forming a scaffold that supports tissue architecture, our data identify that normalising the biomechanical properties of fibrillar collagen in lung fibrosis may reset mechanosensitive cellular mechanisms and aid restoration of tissue function more effectively than approaches that promote extensive collagen degradation, since significant loss of ECM structure is itself pathological, as evidenced by the emphysematous lung.

In summary, this integrated study identifies that altered collagen architecture is a key determinant of abnormal ECM structure-function in human lung fibrosis at time of diagnosis. Dual inhibition of LOXL2 and LOXL3 reduces pathological pyridinoline cross-links, normalises collagen morphology, reduces tissue stiffness, and improves lung function. Thus, targeting pathways which dysregulate collagen architecture may restore ECM homeostasis and so prevent persistent mechanosensitive cellular activation and fibrosis progression.

## Materials and methods

### Lung tissue sampling

All human lung experiments were approved by the Southampton and South West Hampshire and the Mid and South Buckinghamshire Local Research Ethics Committees (ref 07/H0607/73), and all subjects gave written informed consent. Clinically indicated IPF lung biopsy tissue samples and age-matched non-fibrotic control tissue samples (macroscopically normal lung sampled remote from a cancer site in patients undergoing surgery for early stage lung cancer) deemed surplus to clinical diagnostic requirements were flash frozen and stored in liquid nitrogen. All IPF samples were from patients subsequently receiving a multidisciplinary diagnosis of IPF according to international consensus guidelines (*Raghu et al., 2011*). Unless otherwise indicated, serial 50 μm cryosections were utilised for: enzymatic extraction of collagen for AFM-based nanoindentation experiments (five sections), gene expression analysis (five sections), AFM-based microindentation experiments (two sections), hydrolysis for hydroxyproline and collagen cross-link quantification (approximately 1 mm$^3$ of adjacent lung tissue following cryosection). For second harmonic generation imaging archived formalin-fixed paraffin-embedded samples from control and IPF donors were studied.

## Atomic force microscopy (AFM) experiments

AFM-based experiments were performed using a NanoWizard ULTRA Speed A AFM system (JPK Instruments AG, Berlin). For AFM-nanoindentation experiments, pyrex nitride cantilevers (Nano-World AG, Switzerland) of 0.48 Nm$^{-1}$ nominal spring constant were used and for microindentation experiments CSC38 cantilevers (μMasch, Innovative Solutions Bulgaria Ltd., Bulgaria) of 0.03 Nm$^{-1}$ spring constant and a 15 μm diameter microsphere were used. The spring constant is defined as the ratio of the force affecting the spring to the displacement caused by it. During mechanical assessment samples were hydrated in phosphate buffered saline (pH7.4).

AFM-based microindentation experiments were performed on 50 μm sections. By attaching a microsphere on to the end of the flexible AFM cantilever, microindentation enables assessment of the mechanics of larger structures, such as localised areas on tissue sections composed of hundreds of collagen fibrils (*Kain et al., 2018*). The tip radius of these spheres was approximately 7.5 μm, giving a volume interaction between the sphere and the tissue in the order of 10–100 μm$^3$. Frozen cryosections were warmed to room temperature in PBS containing a protease inhibitor cocktail to minimise tissue degradation and between 80 and 150 force-displacement curve measurements were performed at 6 to 11 sites across each tissue section.

For nanoindentation, collagen was enzymatically extracted from tissue sections using 1 mg/ml bovine hyaluronidase and 1 mg/ml trypsin in 0.1 M Sorensen's phosphate buffer (pH 7.2) at 37°C for 24 hr (*Stolz et al., 2009*; *Andriotis et al., 2014*). After thorough washing with deionised water, the samples were smeared on to glass slides to reveal areas with individual fibrils. Samples were then left to dry overnight at 37°C and stored in a desiccating storage box with silica gels until analysis. Extracted collagen fibrils were assessed by employing a conical tip with tip radius of 7–10 nm at the end of a flexible cantilever, giving a volume interaction between the tip and collagen fibril of 0.1–1 nm$^3$. AFM imaging prior to force-displacement acquisition of the dry sample enabled individual collagen fibrils to be identified from the characteristic 67 nm D periodicity. The sample was then hydrated in PBS and force-displacement data were recorded and subsequently analysed with the well-established mathematical Oliver-Pharr model (*Oliver and Pharr, 2004*), so enabling determination of the indentation modulus of the measured collagen fibril (*Andriotis et al., 2014*). For each tissue sample, a minimum of four collagen fibrils were studied, with 30 to 50 force-displacement curves measured per fibril. The total acquisition time was kept as low as possible to minimise thermal drift effects.

## Reverse transcription quantitative polymerase chain reaction (RTqPCR)}

RNA was isolated using a Savant FastPrep FP12 Ribolyser (Qbiogene, Cedex, France) and RNeasy Mini kits (Qiagen, Manchester, UK) according to the manufacturer's instructions. For the in vitro lung fibrosis model, RLT buffer also contained Proteinase K (Sigma, Poole, UK) to aid lysis. RNA was reverse transcribed to cDNA using a QuantiTect Reverse Transcription Kit or Precision nanoScript2 Reverse Transcription Kit (Primer design, Southampton, UK) according to manufacturer's instructions. Real-time quantitative polymerase chain reaction (RTqPCR) was performed using a BioRad CFX96 (BioRad, Watford, UK) or a NanoString (NanoString Technologies, Seattle, USA) system with a custom CodeSet. Primers and TaqMan probe sets were obtained from Primer Design. Changes in gene expression were compared to two or three constitutively expressed housekeeping genes (for human samples *UBC/A2* and for rat samples *Pgk1/Actab/Hprt1*). Data were normalised to the mean of the housekeeping genes, and these data normalised to an appropriate baseline control sample.

## Second harmonic imaging

Formalin-fixed paraffin-embedded human lung tissue sections (5 μm) from control and IPF donors (n = 5 per group) were dewaxed and imaged using a custom laser scanning microscope (output of an Optical Parametric Oscillator (two picosecond pulses, at 80MHz repetition rate Levante Emerald OPO, pumped by Emerald engine, APE, Berlin) coupled into an inverted microscope (Ti-U, Nikon, Japan) via a galvanometric scanner and a short pass excitation dichroic (750 nm cut-off, Chroma 750spxrxt). Samples were excited at 800 nm and the epi-collected second harmonic signal was further filtered with a 442 nm dichroic mirror (Semrock, DI-02-R442) and a band pass filter centred at 400 nm (Thorlabs, FB400-40) and was finally detected with a photomultiplier tube (Hamamatsu 10722–210). Samples were imaged with 20 mW of power measured at the sample using the same

gain for each sample. For image co-registration, an adjacent 5 µm tissue section was H and E stained and imaged using a Dot-Slide scanning system (Olympus, Southend-on-Sea, UK). Image co-registration was performed in the Fiji distribution (*Schindelin et al., 2012*) (version 1.49 p) of ImageJ using landmark correspondences.

### In situ analysis of amine oxidase activity

An in situ assay for the detection of amine oxidase activity in skin (*Langton et al., 2013*) was adapted for use on lung tissue. The assay is based on oxidation of the amine substrate, 1,4-diamino-butane and chemiluminescent detection of liberated $H_2O_2$ with horseradish peroxidase and luminol. Cryosections (5 µm) of non-fibrotic or IPF lung tissue were placed on glass slides and air dried for 10 min. The substrate/detection mixture was applied to each tissue section and incubated at 37°C for 5 min. After removal of excess substrate solution and covering the tissue section with a coverslip, each section was digitally imaged with chemiluminescence detection (Amersham Imager 600, GE Healthcare Life Sciences, Little Chalfont, UK). As a control to verify the contribution of LOX/LOXL to the amine oxidase activity in the tissue, a serial section was preincubated with the irreversible LOX/LOXL inhibitor, BAPN (300 µM) for 30 min prior to substrate addition. After imaging, semi-quantitative analysis was performed, with the mean grey scale value of each tissue area following background subtraction calculated using Fiji (version 1.49 p), with each sample normalised to the wet weight of 250 µm of adjacent cryosections.

### Inhibition profiling of PXS-S2A and PXS-S3B towards the LOX/LOXL family enzymes

Bovine LOX and recombinant LOXL enzymes were tested in an Amplex Red assay using putrescine as substrate over a range of concentrations of inhibitors as previously described (*Schilter et al., 2015*; *Chang et al., 2017*). Putrescine was used at 10 mM for LOX and LOXL1, 5 mM for LOXL2 and 2 mM for LOXL3 and LOXL4 reflecting the different $K_m$ values for each enzyme. Bovine LOX was extracted from calf aorta (*Schilter et al., 2015*), recombinant LOXL2 and LOXL3 proteins were purchased from R and D systems, recombinant LOXL1 was stably expressed as a his-tagged protein in NIH-3T3 cells and purified from conditioned media using Nickel-Sepharose affinity purification; recombinant LOXL4 was partially purified from conditioned medium from HEK-293 stably transfected with human LOXL4 using size exclusion enrichment with Amicon 50 kDa centrifugal filter units. The LOXL4 cell line was provided by Dr Fernando Rodríguez Pascual, Universidad Autónoma de Madrid, Spain. All proteins were initially characterised by western blotting and enzymatic activity, in order to confirm identity and to yield comparable $V_{max}$ across all assays.

### 3D in vitro model of fibrosis

Peripheral lung fibroblasts were obtained as outgrowths from surgical lung biopsy tissue (*Davies et al., 2012*) of patients (n = 3 donors) who were subsequently confirmed with a diagnosis of IPF. All primary cultures were tested and free of mycoplasma contamination. The fibroblasts were seeded in Transwell inserts in DMEM containing 10% FBS. After 24 hr, the media was replaced with DMEM/F12 containing 5% FBS, 10 µg/ml L-ascorbic acid-2-phosphate, 10 ng/ml EGF, and 0.5 µg/ml hydrocortisone with or without inhibitors, as indicated; each experiment included a vehicle control (0.1% DMSO). TGF-$\beta_1$ (3 ng/mL) was added to the cultures, and the medium replenished three times per week. After 6 weeks the spheroids were either harvested into RNA*later* (Sigma) for RNA extraction, snap frozen for parallel-plate compression testing, analysis of cross-linking, and histochemical staining, or fixed using 4% paraformaldehyde for histochemistry or 3% glutaraldehyde in 0.1 M cacodylate buffer at pH 7.4 for electron microscopy.

### Tinctorial stains

4–7 µm sections were analysed using PicroSirius Red (Abcam, Cambridge, UK) or Masson's Trichrome (Sigma, Poole, UK) stain according to the manufacturers' instructions. Images were acquired using a Dot-Slide scanning system (Olympus, Southend-on-Sea, UK) with PicroSirius Red staining visualised under polarised light. PicroSirius Red images were converted to 16-bit and pseudo-coloured through application of the Yellow Lookup Table within Fiji (version 1·49 c).

## Protein, hydroxyproline and collagen cross-link analysis

Samples were thawed and reduced with $KBH_4$ before acid hydrolysis in 6M HCl at 102°C for 18 hr. Cooled hydrolysed samples were evaporated to dryness under vacuum and then resuspended in 200 µL HPLC-grade $H_2O$. Total protein was quantified in the hydrolysed samples using a genipin-based amino acid assay (QuickZyme Biosciences, Leiden, The Netherlands), using standard hydrolysed bovine serum albumin as standard.

Total collagen content was estimated using either ultra-high performance liquid chromatography-electrospray ionisation tandem mass spectrometry (UHPLC-ESI-MS/MS) or by colorimetric assay of hydroxyproline (Hyp) based on the reaction of oxidized hydroxyproline with 4-(Dimethylamino)benzaldehyde, as per manufacturer's instruction (Sigma-Aldrich, Poole, UK). The molar content of collagen was estimated from hydroxyproline using a conversion factor of 300 hydroxyprolines per triple helix, and mass of collagen was estimated using a molecular weight of 300 kDa per triple helix (**Miller et al., 1990**).

Collagen cross-links were determined using either UHPLC-ESI-MS/MS or ELISA as indicated in the figure legends. Total mature pyridinium cross-links (PYD +DPD) were determined using enzyme-linked immunosorbent assay (ELISA; Quidel Corporation, San Diego, USA) according to manufacturer's instructions. For UHPLC-ESI-MS/MS analysis, cross-links were first extracted from the hydrolysate using an automated solid phase extraction system (Gilson GX-271 ASPECA system) employing reversed-phase C18-Aq columns (GracePure, ThermoFisher Scientific, Australia) followed by SCX strong cation exchange columns. After extraction and drying, the cross-links were converted into heptafluorobutyric acid (HFBA) salts for analysis by UHPLC-ESI-MS/MS on a Thermo Dionex UHPLC and TSQ Endura triple quad mass spectrometer. UHPLC separation of cross-links was achieved with an Agilent Rapid Resolution High Definition (RRHD) SB-C18 column. UHPLC was performed using a 12-min gradient flow of the mobile phase A (10 mM ammonium formate, 0.1% formic acid, 0.1% HFBA in $H_2O$) from 96.2% to 0% and mobile phase B (10 mM ammonium formate, 0.1% formic acid, 0.1% HFBA in 80% MeOH) from 3.8% to 100% at a flow rate of 0.3 mL/min and with a column temperature of 40°C. Positive ESI-MS/MS with Selected Reaction Monitoring (SRM) mode was performed using the following parameters: Spray voltage 4000 V; sheath gas 35 (Arb); aux gas 20 (Arb); sweep gas 0 (Arb); ion transfer tube temperature 350°C; vaporiser temperature 300°C. Standards used for quantitation of collagen cross-links or total collagen in hydrolysates by UHPLC-ESI-MS/MS were: DHLNL (Thermo Fisher Scientific, Scoresby, Australia), HLNL (Toronto Research Chemicals, Toronto, Canada), PYD and DPD (Quidel Corporation, San Diego, USA) and hydroxyproline (Sigma-Aldrich, Poole, UK). Quantitation of the collagen cross-links and total collagen was achieved by comparing to a standard curve. Sample values were interpolated using GraphPad Prism seven software.

## Parallel plate compression testing

To determine the stiffness characteristics (Young's modulus, *E*) of the 3D in vitro model of fibrosis, cultures were subjected to parallel plate compression testing using a CellScale MicroSquisher fitted with a round tungsten cantilever (thickness 406.4 nm) and accompanying SquisherJoy V5.23 software (CellScale, Ontario, Canada). The fluid bath test chamber was filled with sterile PBS, and the stage and optics calibrated following the manufacturer's instructions. Example test sequences are shown in *Figure 6—figure supplement 1* and *Video 1*. Samples were subjected to five testing cycles under hydrated conditions, each achieving 25% engineering strain (deformation) over a 15 s compression phase, followed by a 2 s hold, recovery over 15 s, and a 2 s resting period. The first four cycles allowed preconditioning of the tissue, and the resulting force vs time curves could be seen to have stabilised by the fifth compression cycle (*Figure 6—figure supplement 1a*). Analysis of stress vs strain relationships was carried out using the compression phase of the fifth cycle from where sample stiffness can be inferred. Here, we restrict ourselves to the estimation of the Young's modulus of the biological material in hydrated state.

Force and displacement data were transformed to engineering stress versus engineering strain plots using the horizontal cross-sectional diameter of the sample immediately before the start of each test. Young's modulus (*E*), a measure of stiffness, was calculated using a modified Hertzian half-space contact mechanics model for elastic spheres as previously described (Kim et al., 2010), averaging between values of *E* calculated for both the horizontal and vertical radii of the quasi-spherical cultures. This calculation requires the value of the Poisson's ratio $\nu$. Given the narrow range in which

this value lies, the relative insensitivity of the calculations to this material parameter, and significant difficulty in their direct measurement for small biological samples, here we make use of estimated values taken as 0.5, which is realistic for a range of biological matter. All stress-strain curves show an initial 'toe' region, where the stress increases slowly, followed by a linear region, approximately between 10% and 20% strain. The stiffness of the tissue was obtained by averaging the computed values of $E$ within this linear range.

## Transmission electron microscopy

Samples were post-fixed sequentially in osmium/ferrocyanide fixative, thiocarbohydrazide solution, osmium tetroxide, uranyl acetate and Walton's lead aspartate solution before dehydration in graded ethanol and acetonitrile. Samples were embedded in Spurr resin and 100 nm ultra-thin sections visualised using an FEI Tecnai 12 transmission electron microscope (FEI Company, Hillsboro, OR, USA). For measurement of fibril diameters images were acquired at 87,000x magnification. Measurements of fibril diameter from two independent experiments were made by two operators in parallel blinded to the treatment conditions using ImageJ (*Schneider et al., 2012*) (Version 1.50 g). The shortest axis of each fibril was measured with a total of 300 measurements per sample.

## Collagen matrix solubility assay

To assess collagen solubility samples from the 3D in vitro model of lung fibrosis were subjected to sequential collagen extraction under increasingly stringent conditions as described previously (*Liu et al., 2016*). Briefly, samples were subjected to sequential overnight incubations in 1) neutral salt (Tris buffered saline containing protease inhibitors); 2) 0.5M acetic acid with protease inhibitors; and 3) 0.5M acetic acid containing 700 units/mL pepsin (Sigma, Poole, UK). Each fraction, and the remaining solid (insoluble highly cross-linked) fraction, was hydrolysed in 6M HCl at 102°C for 18 hr and the collagen content of each fraction determined by colorimetric hydroxyproline assay.

## Adenovirus TGF-β model

Studies in rats were performed at McMaster University (Hamilton, Ontario, Canada) according to guidelines from the Canadian Council on Animal Care and approved by the Animal Research Ethics Board of McMaster University under protocol # 16-04-14. Sample sizes were based on previous studies in the laboratory where the studies were performed and were chosen to balance the ability to measure significant differences while reducing the number of animals used. Female Sprague Dawley rats (Charles River Laboratories, Montreal, Canada) were randomly assigned to receive $2.5 \times 10^8$ pfu of either a replication-deficient adenoviral vector carrying an active TGF-β1 (AdTGF-β1) expression construct to induce progressive pulmonary fibrosis (*Sime et al., 1997*) or vector only control (AdDL) on Day 0. Treatment groups received LOXL2/LOXL3 inhibitor (15 or 30 mg/kg/day dosed daily via the oral route) from day 1 to day 28; control groups received vehicle (PBS). On day 28, lung function was assessed using forced oscillation manoeuvres with a FlexiVent system (SCIREQ Scientific Respiratory Equipment, Montreal, Canada) to measure pressure-volume loops, resistance (R), compliance (C), and elastance (E) of the respiratory system. Animals were then euthanised and the left lung lobe inflated under constant pressure and fixed in 10% formalin for histological examination of tissue sections by staining with picrosirius red and viewing under polarised light. Fibrosis was assessed by six independent, trained individuals who were blinded to the treatment conditions using a modified Ashcroft score (*Bonniaud et al., 2005*). All right lung lobes were combined, flash frozen and ground into a fine powder under liquid nitrogen and stored for quantitation of collagen and its cross links, and for RNA analysis.

## Statistics

Statistical analyses were performed in GraphPad Prism v7.02 (GraphPad Software Inc., San Diego, CA) unless otherwise indicated. No data were excluded from the studies and for all experiments, all attempts at replication were successful. For each experiment, sample size reflects the number of independent biological replicates and is provided in the figure legend. Normality of distribution was assessed using the D'Agostino-Pearson normality test. Statistical analyses of single comparisons of two groups utilised Student's *t*-test or Mann-Whitney U-test for parametric and non-parametric data respectively. Where appropriate, individual *t*-test results were corrected for multiple comparisons

using the Holm-Sidak method. For multiple comparisons, one-way analysis of variance (ANOVA) with Dunnett's multiple comparison test or Kruskal-Wallis analysis with Dunn's multiple comparison test were used for parametric and non-parametric data, respectively. In the in vitro studies, repeated measures (RM) ANOVA was used in order to treat data from each donor as a matched set. Sphericity of the repeated measures was assumed since the experiments were designed as randomised blocks. Multivariate correlations were performed using JMP software v13.1.0 (SAS, Marlow, UK) applying the restricted maximum likelihood (REML) method to handle missing data. Results were considered significant if $p < 0.05$, where $*p < 0.05$, $**p < 0.01$, $***p < 0.001$, $****p < 0.0001$.

## Acknowledgements

This study was supported by a Wellcome Trust Clinical Research Training Fellowship (100638/Z/12/Z), the Medical Research Council (MRC, Grant number G0900453), the Medical Research Foundation (MRF-091–0003-RG-CONFO), the British Lung Foundation (RG14-4 and IPFPG12-2), and the Canadian Institutes for Health Research (MOP-136950). Infrastructure support was provided by the National Institute for Health Research (NIHR) Southampton Respiratory Biomedical Research Unit and Wessex Clinical Research Network. We thank the staff of the Biomedical Imaging Unit and the Histochemistry Research Unit at University Hospital Southampton. We acknowledge the support of the NIHR through an Academic Clinical Fellowship and Academic Clinical Lectureship. We acknowledge the support of the Biolab Instrumentation suite and an EPSRC Doctoral Prize (to A. Bonfanti). SM acknowledges support through the European Research Council (ERC, Grant number 638258), and TS support though a C Jane and C Robert Davis Endowed Professorship. The authors would like to thank Dr Kevin Eliceiri and Prof Paul Campagnola of the Univeristy of Madison-Wisconsin, USA for advice on SHG imaging, and Prof Simon Robins of the University of Aberdeen, UK for advice on measurement of collagen cross-linking.

## Additional information

### Competing interests

James JW Roberts: As a current employee of Synairgen Research Ltd, J.J.W.R has share options in the company. Kerry Lunn: As a current employee of Synairgen Research Ltd, K.L has share options in the company. Victoria J Tear: As a current employee of Synairgen Research Ltd, V.J.T has share options in the company. Lucy Cao: As a former or current employee of Pharmaxis Pharmaceuticals Ltd, L.C. has an equity stake in the company. Wolfgang Jarolimek: As a current employee of Pharmaxis Pharmaceuticals Ltd, W.J. has an equity stake in the company. Phillip D Monk: As a current employee of Synairgen Research Ltd, P.D.M has share options and is a share-holder in the company. Donna E Davies: D.E.D is co-founder of, share-holder in and consultant to Synairgen Research Ltd. The other authors declare that no competing interests exist.

### Funding

| Funder | Grant reference number | Author |
|---|---|---|
| Wellcome | 100638/Z/12/Z | Mark G Jones |
| Canadian Institutes of Health Research | MOP-136950 | Kjetil Ask<br>Jack Gauldie<br>Martin Kolb |
| Medical Research Foundation | | Franco Conforti |
| H2020 European Research Council | 638258 | Sumeet Mahajan |
| Medical Research Council | G0900453 | Philipp J Thurner<br>Donna E Davies |
| British Lung Foundation | RG14-4 | Donna E Davies |
| British Lung Foundation | IPFPG12-2 | Donna E Davies |

The funders had no role in study design, data collection and interpretation, or the decision to submit the work for publication.

## Author contributions

Mark G Jones, Conceptualization, Data curation, Formal analysis, Supervision, Funding acquisition, Investigation, Methodology, Writing—original draft, Project administration, Writing—review and editing; Orestis G Andriotis, Kerry Lunn, Victoria J Tear, Kjetil Ask, David E Smart, Formal analysis, Investigation, Methodology, Writing—review and editing; James JW Roberts, Formal analysis, Investigation, Methodology, Writing—original draft, Writing—review and editing; Lucy Cao, Formal analysis, Investigation, Methodology; Alessandra Bonfanti, Formal analysis, Writing—review and editing; Peter Johnson, Christopher J Brereton, Investigation, Writing—review and editing; Aiman Alzetani, Chester Y Lai, Konstantinos N Bourdakos, Sanjay Jogai, Patricia Sime, Resources, Methodology, Writing—review and editing; Franco Conforti, Wolfgang Jarolimek, Resources, Investigation, Methodology, Writing—review and editing; Regan Doherty, Sumeet Mahajan, Aurelie Fabre, Investigation, Methodology, Writing—review and editing; Benjamin Johnson, Sophie V Fletcher, Resources, Writing—review and editing; Ben G Marshall, Serena J Chee, Christian H Ottensmeier, Resources, Project administration, Writing—review and editing; Jack Gauldie, Martin Kolb, Formal analysis, Investigation, Writing—review and editing; Atul Bhaskar, Formal analysis, Methodology, Writing—review and editing; Luca Richeldi, Conceptualization, Formal analysis, Writing—review and editing; Katherine MA O'Reilly, Conceptualization, Formal analysis, Methodology, Writing—review and editing; Phillip D Monk, Conceptualization, Resources, Formal analysis, Supervision, Investigation, Methodology, Project administration, Writing—review and editing; Philipp J Thurner, Conceptualization, Resources, Formal analysis, Supervision, Funding acquisition, Investigation, Methodology, Writing—review and editing; Donna E Davies, Conceptualization, Resources, Formal analysis, Supervision, Funding acquisition, Investigation, Methodology, Writing—original draft, Project administration, Writing—review and editing

## Author ORCIDs

Mark G Jones https://orcid.org/0000-0001-6308-6014
Christopher J Brereton http://orcid.org/0000-0001-8302-702X
Christian H Ottensmeier http://orcid.org/0000-0003-3619-1657
Donna E Davies https://orcid.org/0000-0002-5117-2991

## Ethics

Human subjects: All human lung experiments were approved by the Southampton and South West Hampshire and the Mid and South Buckinghamshire Local Research Ethics Committees (ref 07/H0607/73), and all subjects gave written informed consent.

Animal experimentation: Studies in rats were performed at McMaster University (Hamilton, Ontario, Canada) according to guidelines from the Canadian Council on Animal Care and approved by the Animal Research Ethics Board of McMaster University under protocol # 16-04-14.

## Decision letter and Author response

Decision letter https://doi.org/10.7554/eLife.36354.017
Author response https://doi.org/10.7554/eLife.36354.018

# Additional files

## Supplementary files

• Transparent reporting form
DOI: https://doi.org/10.7554/eLife.36354.015

## Data availability

All data generated or analysed during this study are included in the manuscript and supporting files.

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
