## [Decision Letter]

Thank you for submitting your article "Nanoscale dysregulation of collagen structure-function disrupts mechano-homeostasis and mediates pulmonary fibrosis" for consideration by *eLife*. Your article has been reviewed by three peer reviewers, and the evaluation has been overseen by a Reviewing Editor and Randy Schekman as the Senior Editor. The following individual involved in review of your submission has agreed to reveal his identity: Bela Suki (Reviewer #1).

The reviewers have discussed the reviews with one another and the Reviewing Editor has drafted this decision to help you prepare a revised submission.

Summary:

This is an excellent and very interesting manuscript that shows that the collagen structure and its mature cross-linking, rather than the collagen quantity, change during pulmonary fibrosis, leading to the disruption of mechanobiological homeostasis of the lung. It is clearly demonstrated that the fibrotic lung matrix is stiffer than normal lung matrix, and that this stiffness is secondary to changes in collagen nanostructure rather than in the collagen content. Although some of the conclusions reported here have been reached by some other authors using animal models, this manuscript is a more comprehensive study, with strong focus on the human lung. Notably, the authors also show that the selective inhibition of one set of LOX genes, LOXL2/LOXL3 may provide therapeutic benefits. The studies are rigorous and will be of major interest to the field.

The reviewers also raised some concerns that can be addressed through revisions of the manuscript, and in one instance by including some additional experimental data. We would encourage the authors to address these comments and further increase the already high value and impact of this manuscript.

Essential revisions:

1) It is intuitively difficult to see how there cannot be a difference in total collagen content, given the severely thickened interstitium in fibrotic lung tissue. Please clarify if this thickening can be fully explained by accumulation of mesenchymal cells, and if the hydroxyproline content is the only way (or the right way) to assess the collagen contents in this case. Also, It would be useful to better reconcile some of the data obtained here with previous publications that show clear evidence of increased collagen deposition.

2) The major claim of the study is that it is not collagen deposition, but cross-linking that really matters in lung fibrosis. Related to this claim, total collagen concentration in Figure 1E showed no difference between normal and IPF lungs. However, the cellular behavior is affected by the what could matter is the local concentration of collagen, and the images in Figure 1 indicate that there is a heterogeneity in collagen density of collagen between different areas of the lung. Apparently, the AFM measurements were also taken in a single location. The heterogeneity of the diseased lung is an important factor, possibly even more important that the measured mean values. The authors should provide additional measurements that would address the issue of heterogeneity of the lung composition and stiffness.

3) The authors downplayed the importance of immature as opposed to mature crosslinks on tissue stiffness. In Figure 3, it's surprising that the mature cross-links are so strongly associated with stiffness. Please explain and document if there is a relationship between stiffness and DHLNL alone. In Figure 5 I, J, immature crosslinks appear more important contributors to stiffness than the authors acknowledge in the text.

4) Another important issue is collagen hydration, as it has large influence on both collagen stiffness (Yang et al., 2008) and collagen fibril diameter up to a max diameter after which it shows a plateau (Fratzl and Daxer. Biophys J 1993, 64, (4), 1210-4). The stress-strain curves of hydrated and dry fibrils are very different and the mechanism of molecular slippage and stretching within the fibrils depend on the hydration of the fibril (Gautieri et al., 2011). Thus, it is possible that the fibrotic lungs are dry compared to the normal lung which, next to cross-linking, also contributes significantly to the difference in stiffness. Notably, cross-linking could change collagen hydration and thereby have important contribution to stiffness. Even though hydration is touched upon in the discussion, it should be addressed in more detail.

---

## [Author Response]

Essential revisions:1) It is intuitively difficult to see how there cannot be a difference in total collagen content, given the severely thickened interstitium in fibrotic lung tissue. Please clarify if this thickening can be fully explained by accumulation of mesenchymal cells, and if the hydroxyproline content is the only way (or the right way) to assess the collagen contents in this case. Also, It would be useful to better reconcile some of the data obtained here with previous publications that show clear evidence of increased collagen deposition.

In fibrotic lung tissue there are marked histological abnormalities including thickened interstitium. Whilst visualisation of a 2D histological section through immunohistochemical or second harmonic generation imaging may identify an increase in collagen in these areas, it is not able to identify whether this increase is in isolation, or proportional to other ECM and cellular components i.e. the absolute amount of collagen may increase without any increase in the relative amount. We applied hydroxyproline as it is a reference standard for biochemical quantitation of all forms of collagen irrespective of whether it is assembled into fibrils. We identified that total collagen relative to dry weight or total protein (i.e. in proportion to other lung constituents) is not increased at the time of diagnosis. Consistent with our findings, the largest comparable study (Kirk et al., 1986) identified no increase in collagen concentration assessed by hydroxyproline in biopsy samples vs control samples. Only in autopsy samples was an increase in collagen concentration identified. This suggests that any long-term increase in collagen concentration is a downstream event is of less pathogenetic relevance than the changes in cross-linking and structure we have identified to be present at time of diagnosis. We have expanded our Discussion to further discuss these points (second paragraph).

2) The major claim of the study is that it is not collagen deposition, but cross-linking that really matters in lung fibrosis. Related to this claim, total collagen concentration in Figure 1E showed no difference between normal and IPF lungs. However, the cellular behavior is affected by the what could matter is the local concentration of collagen, and the images in Figure 1 indicate that there is a heterogeneity in collagen density of collagen between different areas of the lung. Apparently, the AFM measurements were also taken in a single location. The heterogeneity of the diseased lung is an important factor, possibly even more important that the measured mean values. The authors should provide additional measurements that would address the issue of heterogeneity of the lung composition and stiffness.

In lung fibrosis there is marked histological heterogeneity over the micrometre scale in tissue sections (usually 3-4 μm thick). The AFM analyses were performed on 50 μm thick sections, with 80 to 150 force-displacement curve measurements taken at 6 to 11 sites across the tissue sample, and so the measurements are representative of the whole tissue sample studied. We have included these specific details in the Materials and methods (subsection “Atomic Force Microscopy (AFM) Experiments”, second paragraph). Whilst IPF tissue is heterogeneous, our in vitro model of fibrosis is homogeneous, and our data are consistent with our tissue findings: thus, within the model (Figure 6H) collagen concentration did not influence tissue stiffness whilst cross-linking did. To directly investigate the heterogeneity of lung composition and stiffness at a cellular level would require consideration of multiple factors, not only those influencing collagen mechanics (collagen cross-linking, concentration, topology, hydration) at that site but also the presence of other ECM proteins and the cells themselves. This would require significant methodological development to enable co-registration of these multiple readouts at the μm scale which is beyond our current capabilities.

3) The authors downplayed the importance of immature as opposed to mature crosslinks on tissue stiffness. In Figure 3, it's surprising that the mature cross-links are so strongly associated with stiffness. Please explain and document if there is a relationship between stiffness and DHLNL alone. In Figure 5 I, J, immature crosslinks appear more important contributors to stiffness than the authors acknowledge in the text.

It was not our intention to downplay the importance of immature crosslinks on tissue stiffness, however within our human tissue analyses no significant association of immature crosslinks with tissue stiffness was identified, and so we focussed upon mature crosslinks. As requested by the reviewer, we examined the relationship with DHLNL cross-links alone in our tissue analysis and found a trend towards a correlation however this was not statistically significant (correlation 0.407, p=0.09); with HLNL no correlation was observed (correlation 0.242, p=0.33). We have included a statement describing this within the results (subsection “Increased mature collagen cross-linking but not total collagen content increases stiffness of IPF tissue”, last paragraph). Within our in vitro analyses a relationship with immature cross-links was identified, although that with mature cross-links was stronger. We have now directly commented upon this contribution of immature cross-links (subsection “LOXL2/LOXL3-selective inhibition of collagen cross-linking reduces tissue stiffness”, last paragraph).

4) Another important issue is collagen hydration, as it has large influence on both collagen stiffness (Yang et al., 2008) and collagen fibril diameter up to a max diameter after which it shows a plateau (Fratzl and Daxer. Biophys J 1993, 64, (4), 1210-4). The stress-strain curves of hydrated and dry fibrils are very different and the mechanism of molecular slippage and stretching within the fibrils depend on the hydration of the fibril (Gautieri et al., 2011). Thus, it is possible that the fibrotic lungs are dry compared to the normal lung which, next to cross-linking, also contributes significantly to the difference in stiffness. Notably, cross-linking could change collagen hydration and thereby have important contribution to stiffness. Even though hydration is touched upon in the discussion, it should be addressed in more detail.

We thank the reviewers for requesting us to further highlight the role of collagen hydration upon collagen fibril mechanics, and we have expanded our discussion of this factor accordingly (Discussion, third paragraph). It is an interesting idea that fibrotic lung tissue is drier than normal lung. It is notable that in IPF the sites of active fibrosis have multiple ECM components with high affinity for water such as hydrophilic proteoglycans (Bensadoun et al. Am J Respir Crit Care Med 1996;154:1819-1828). Conceivably these could compete locally with the collagen fibrils for water molecules to further affect fibril stiffness.